# Certified Defences Against Adversarial Patch Attacks on Semantic Segmentation

**Maksym Yatsura**
Bosch Center for Artificial Intelligence
University of Tübingen
`maksym.yatsura@de.bosch.com`

**Kaspar Sakmann**
Bosch Center for Artificial Intelligence
`kaspar.sakmann@de.bosch.com`

**N. Grace Hua**
Bosch Center for Artificial Intelligence
`grace.hua@de.bosch.com`

**Matthias Hein**
University of Tübingen
`matthias.hein@uni-tuebingen.de`

**Jan Hendrik Metzen**
Bosch Center for Artificial Intelligence
`janhendrik.metzen@de.bosch.com`

## Abstract

Adversarial patch attacks are an emerging security threat for real world deep learning applications. We present DEMASKED SMOOTHING, the first approach (up to our knowledge) to certify the robustness of semantic segmentation models against this threat model. Previous work on certifiably defending against patch attacks has mostly focused on image classification task and often required changes in the model architecture and additional training which is undesirable and computationally expensive. In DEMASKED SMOOTHING, any segmentation model can be applied without particular training, fine-tuning, or restriction of the architecture. Using different masking strategies, DEMASKED SMOOTHING can be applied both for certified detection and certified recovery. In extensive experiments we show that DEMASKED SMOOTHING can on average certify 63% of the pixel predictions for a 1% patch in the detection task and 46% against a 0.5% patch for the recovery task on the ADE20K dataset.

## 1 Introduction

Physically realizable adversarial attacks are a threat for safety-critical (semi-)autonomous systems such as self-driving cars or robots. Adversarial patches [1, 2] are the most prominent example of such an attack. Their realizability has been demonstrated repeatedly, for instance by Lee and Kolter [3]: an attacker places a printed version of an adversarial patch in the physical world to fool a deep learning system. While empirical defenses [4–7] may offer robustness against known attacks, it does not provide any guarantees against unknown future attacks [8]. Thus, certified defenses for the patch threat model, which allow guaranteed robustness against all possible attacks for the given threat model, are crucial for safety-critical applications.

Research on certifiable defenses against adversarial patches can be broadly categorized into certified recovery and certified detection. *Certified recovery* [8–16] has the objective to make a correct prediction on an input even in the presence of an adversarial patch. In contrast, *certified detection* [17–20] provides a weaker guarantee by only aiming at *detecting* inputs containing adversarial patches. While certified recovery is more desirable in principle, it typically comes at a high cost of

2022 Trustworthy and Socially Responsible Machine Learning (TSRML 2022) co-located with NeurIPS 2022.

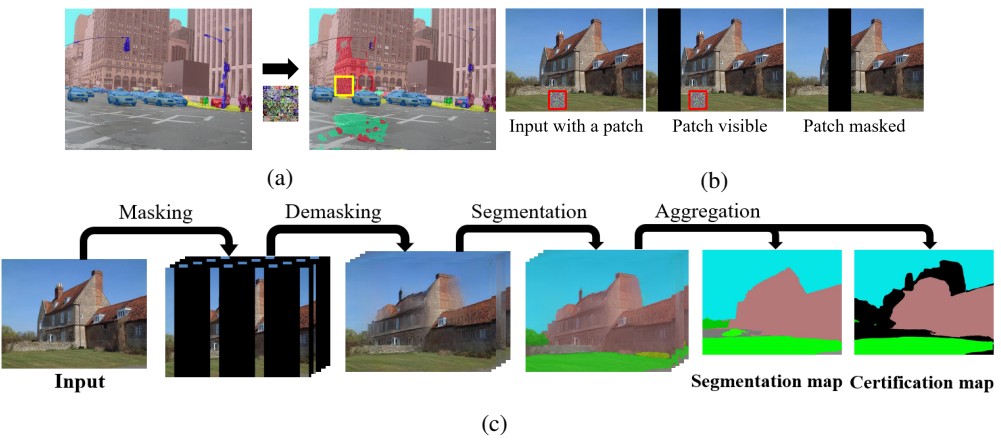

(a)                                                        (b)

Masking          Demasking          Segmentation          Aggregation

Input                                              Segmentation map  Certification map

(c)

Figure 1: (a) A simple patch attack on the Swin transformer [26] manages to switch the prediction for a big part of the image. (b) Masking the patch. (c) A sketch of DEMASKED SMOOTHING for certified image segmentation. First, we generate a set of masked versions of the image such that each possible patch can only affect a certain number of masked images. Then we use image inpainting to partially recover the information lost during masking and then apply an arbitrary segmentation method. The output is obtained by aggregating the segmentations pixelwise.

reduced performance on clean data. In practice, certified detection might be preferable because it allows maintaining high clean performance. Most existing certifiable defenses against patches are focused on image classification. DetectorGuard [21] and ObjectSeeker [22] that certifiably defend against patch hiding attacks on object detectors. Moreover, existing defences are not easily applicable to arbitrary downstream models, because they assume either that the downstream model is trained explicitly for being certifiably robust [9, 12], or that the model has a certain network architecture such as BagNet [10, 12, 11] or a vision transformer [15, 20]. PatchCleanser [14], which can be combined with arbitrary downstream models but is restricted to image classification. Adversarial patch attacks were also proposed for the image segmentation problem [23], mostly for attacking CNN-based models that use a localized receptive field [24]. However, recently self-attention based vision transformers [25] have achieved new state-of-the-art in the image segmentation task [26, 27]. Their output may become more vulnerable to adversarial patches if they manage to manipulate the global self-attention [28]. We demonstrate how significant parts of the segmentation output may be affected by a small patch for Swin transformer [26] in Figure 1a (see details in Appendix E). We point out that preventive certified defences are important because newly developed attacks can immediately be used to compromise safety-critical applications unless they are properly defended.

In this work, we propose the novel framework DEMASKED SMOOTHING (Figure 1c) to obtain the first (up to our knowledge) certified defences against patch attacks on semantic segmentation. Similarly to previous work [9], we mask different parts of the input (Figure 1b) and provide robustness guarantees. While prior work required the classification model to deal with such masked inputs, we leverage recent progress in image inpainting [29] to reconstruct the input *before* passing it to the downstream model. This decoupling allows us to support arbitrary downstream models. Moreover, we can leverage state of the art methods for image inpainting. We also propose different masking schemes tailored for the segmentation task that provide the dense input allowing the demasking model to understand the scene but still satisfy the guarantees with respect to the adversarial patch. We summarize our contributions as follows:

- We propose DEMASKED SMOOTHING which is the first (to the best of our knowledge) certified recovery or certified detection based defence against adversarial patch attacks on semantic segmentation models (Section 4).
- DEMASKED SMOOTHING can do certified detection and recovery with any off-the-shelf segmentation model without requiring finetuning or any other adaptation.
- We implement DEMASKED SMOOTHING, evaluate it for different certification objectives and masking schemes (Section 5). We can certify 63% of all pixels in certified detection for a 1% patch and 46% in certified recovery for a 0.5% patch for the BEiT-B [30] segmentation model on the ADE20K [31] dataset.

## 2 Related Work

**Certified recovery.** The first certified recovery defence against patches was proposed by Chiang et al. [8] for classification models . De-Randomized Smoothing (DRS) [9] significantly improved certified accuracy. Models with small receptive fields such as *BagNets* [32] were adopted for this task either by combining them with some fixed postprocessing [10, 11] or by training them end-to-end for certified recovery [12]. DRS was also applied [15] to *Vision Transfomers (ViTs)* [25]. In contrast to these works, our Demasked Smoothing can be applied to models with arbitrary architecture. PatchCleanser [14] has this property as well but it is limited to image classification. Certified recovery against patches has also been extended to object detection to defend against patch hiding attacks [18, 22]. Randomized smoothing [33] has been applied to certify semantic segmentation models against $\ell_2$-norm bounded adversarial attacks [34]. However, to the best of our knowledge, no certified defence against patch attacks for semantic segmentation has been proposed so far.

**Certified detection.** In this alternative to certified recovery, an adversarial patch is allowed to change the model prediction. However, if it succeeds in doing so, the attack is certifiably detected. Minority Reports [17] was the first certified detection method against patches. PatchGuard++ [18] is has significantly improved the inference time. ScaleCert [19] uses "superficial important neurons" to datect an attack. Lastly, PatchVeto [20] implements masking by removing certain input patches of the ViT. In this work, we propose a novel method for certified detection in the semantic segmentation.

**Image reconstruction.** The problem of learning to reconstruct the full image from masked inputs was pioneered by Vincent et al. [35]. It recently attracted attention as proxy task for self-supervised pre-training, especially for the ViTs [30, 36]. Recent approaches to this problem are using Fourier convolutions [37] and ViTs [29]. SPG-Net [38] trains a subnetwork to reconstruct the full semantic segmentation from the masked input.

## 3 Problem Setup

**Semantic segmentation**. In this work, we focus on the semantic segmentation task. Let $\mathcal{X}$ be a set of rectangular images. Let $x \in \mathcal{X}$ be an image with height $H$, width $W$ and the number of channels $C$. We denote $\mathcal{Y}$ to be a finite label set. The goal is to find the segmentation map $s \in \mathcal{Y}^{H \times W}$ for $x$. For each pixel $x_{i,j}$, the corresponding label $s_{i,j}$ denotes the class of the object to which $x_{i,j}$ belongs. We denote $\mathbb{S}$ to be a set of segmentation maps and $f : \mathcal{X} \to \mathbb{S}$ to be a segmentation model.

**Threat model**. Let us consider an untargeted adversarial patch attack on a segmentation model. Consider an image $x \in [0,1]^{H \times W \times C}$ and its ground truth segmentation map $s$. Assume that the attacker can modify an arbitrary rectangular region of the image $x$ which has a size of $H' \times W'$. We refer to this modification as a *patch*. Let $l \in \{0, 1\}^{H \times W}$ be a binary mask that defines the patch location in the image in which ones denote the pixels belonging to the patch. Let $\mathcal{L}$ be a set of all possible patch locations for a given image $x$. Let $p \in [0, 1]^{H \times W \times C}$ be the modification itself. We define an operator $A(x, p, l) = (1 - l) \odot x + l \odot p$, where $\odot$ is element-wise product. The operator $A$ applies the $H' \times W'$ subregion of $p$ defined by a binary mask $l$ to the image $x$ while keeping the rest of the image unchanged. We denote $\mathcal{P} := [0, 1]^{H \times W \times C} \times \mathcal{L}$ to be a set of all possible patch configurations $(p, l)$ that define an $H' \times W'$ patch. Let $s \in \mathbb{S}$ be the ground truth segmentation for $x$ and $Q(f(x), s)$ be some quality metric. The attacker's goal is to find $(p^\star, l^\star) = \arg\min_{(p,\ l) \in \mathcal{P}} Q(f(A(x, p, l)), s)$. In this paper, we propose *certified* defences against *any possible attack* from $\mathcal{P}$ including $(p^\star, l^\star)$. We consider two robustness objectives.

**Certified recovery.** For a pixel $x_{i,j}$ our goal is to verify that the following statement is true

$$\forall (p,\ l) \in \mathcal{P} : \ f(A(x,\ p,\ l))_{i,j} = f(x)_{i,j} \tag{1}$$

**Certified detection.** We define a verification function $v : \mathcal{X} \to \{0,1\}^{H \times W}$. If $v(x)_{i,j} = 1$, then the adversarial patch attack on $x_{i,j}$ can be detected by applying $v$ to the attacked image $x' = A(x, p, l)$.

$$v(x)_{i,j} = 1 \Rightarrow \left[ \forall (p,\ l) \in \mathcal{P} : v(A(x, p, l))_{i,j} = 1 \to f(A(x,\ p,\ l))_{i,j} = f(x)_{i,j} \right] \tag{2}$$

$v(x')_{i,j} = 0$ means an alert on pixel $x'_{i,j}$. However, if $x'$ is not an adversarial example, then this is a false alert. In that case the fraction of pixels for which we return false alert is called *false alert ratio* (FAR). The secondary objective is to keep FAR as small as possible.

Depending on the objective our goal is to certify one of the conditions 1, 2 for each pixel $x_{i,j}$. This provides us an upper bound on an attacker's effectiveness under any adversarial patch attack from $\mathcal{P}$.

# 4 Demasked Smoothing

DEMASKED SMOOTHING (Figure 1c) consists of several steps. First, we apply a predefined set of masks with specific properties to the input image to obtain a set of masked images. Then we reconstruct the masked regions of each image based on the available information with an inpainting model $g$. After that we apply a segmentation model $f$ to the demasked results. Finally, we aggregate the segmentation outcomes and make a conclusion for the original image with respect to the statements (1) or (2). See Algorithm 1 in Appendix B.

## 4.1 Input masking

**Motivation.** Like in previous work (Section 2) we apply masking patterns to the input image and use predictions on masked images to aggregate the robust result. If an adversarial patch is completely masked, it has no effect on further processing. However, in semantic segmentation, we predict not a single whole-image label like in the classification task, but a separate label for each pixel. Thus, making prediction on a masked image must allow us to predict the labels also for the masked pixels.

**Preliminaries.** Consider an image $x \in [0, 1]^{H \times W \times C}$. We define "$*$" to be a special masking symbol that does not correspond to any pixel value and has the property $\forall z \in \mathbb{R}: z \times * = *$. Note that $*$ needs to be different from 0 since 0 is a valid pixel value in unmasked inputs. Let $m \in \{*, 1\}^{H \times W}$ be a *mask*. We call the element-wise product $x \odot m$ a *masking* of $x$. In a masking, a subset of pixels becomes $*$ and the rest remains unchanged. We consider the patches of size at most $H' \times W'$.

**Certified recovery.** We break $m$ into an array $B$ of non-intersecting blocks, each having the same size $H' \times W'$ as the adversarial patch. We index the blocks as $B[q, r], 1 \le q \le \lceil \frac{H}{H'} \rceil, 1 \le r \le \lceil \frac{W}{W'} \rceil$. We say that the block $B[q, r]$ is *visible* in a mask $m$ if $\forall (i, j) \in B[q, r]: m_{i,j} = 1$. Consider an array $M$ of $K$ masks. We define each mask $M[k]$ by a set of blocks that are visible in it. Each block is visible in exactly one mask and masked in the others. We say that a mask $m$ is *affected* by a patch $(p, l)$ if $A(x, p, l) \odot m \ne x \odot m$. We define $T(M) = \max_{(p,l) \in \mathcal{P}} |\{m \in M \,|\, A(x, p, l) \odot m \ne x \odot m\}|$. That is: $T(M)$ is the largest number of masks affected by some patch. If $M$ is defined, we refer to the value $T(M)$ as $T$ for simplicity. We define column masking $M$ for which $T = 2$. We assign every $k$-th block column to be visible in the mask $M[k]$ (Figure 2b). Any $(p, l) \in \mathcal{P}$ can intersect at most two adjacent columns since $(p, l)$ has the same width as a column. Thus, it can affect at most two masks (Figure 2b). A similar scheme can be proposed for the rows. Due to the block size the patch $(p, l)$ cannot intersect more than four blocks at once. We define a mask set that we call *3-mask* s. t. for any four adjacent blocks two are visible in the same mask (Figures 2c). Hence, a patch for 3-mask can affect no more than 3 masks, $T = 3$. To achieve $T = 4$ any assignment of visible blocks to the masks works. We consider *4-mask* that allows uniform coverage of the visible blocks in the image (Figure 2d). See details in Appendix B.

**Certified detection.** We define $M_d$ to be a set of masks for certified detection (we use subscript $d$ for distinction). $M_d$ should have the property: $\forall (p, l) \in \mathcal{P} \, \exists m \in M_d : A(x, p, l) \odot m = x \odot m$ i. e. for every patch exists at least one mask not affected by this patch. See details in Appendix B.

## 4.2 Certification

**Certified recovery.** For the threat model $\mathcal{P}$ consider a set $M$ of $K$ masks. We define a function $h : \mathcal{X} \to \mathbb{S}$ that assigns a class to the pixel $x_{i,j}$ via majority voting over class predictions of each

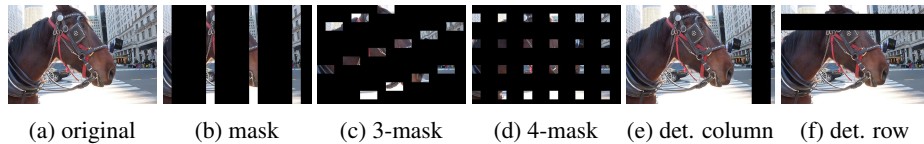

| (a) original | (b) mask | (c) 3-mask | (d) 4-mask | (e) det. column | (f) det. row |

Figure 2: certified recovery: column mask (b), 3-mask (c), 4-mask (d); certified detection (e, f).

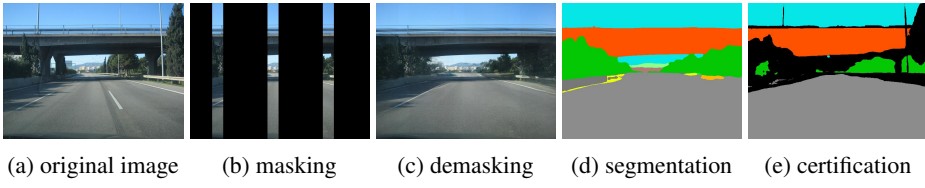

| (a) original image | (b) masking | (c) demasking | (d) segmentation | (e) certification |

Figure 3: Reconstructing the masked images with ZITS [29]

reconstructed segmentation in $S$. A class for the pixel that is predicted by the largest number of segmentations is assigned. We break the ties by assigning a class with a smaller index.

**Theorem 1.** *If the number of masks $K$ satisfies $K \geq 2T(M) + 1$ and for a pixel $x_{i,j}$ we have*

$$\forall \, S[k] \in S : \; S[k]_{i,j} = h(x)_{i,j}$$

*(i.e. all the votes agree), then $\forall \, (p, \, l) \in \mathcal{P} : \; h(A(x, p, l))_{i,j} = h(x)_{i,j}$.*

**Certified detection.** Consider $M_d = \{M_d[k]\}_{k=1}^K$. For a set of demasked segmentations S we define the verification map $v(x)_{i,j} := [f(x)_{i,j} = S[1]_{i,j} = \ldots = S[K]_{i,j}]$ i.e. the original segmentation is equal to all the other segmentations on masked-demasked inputs, including the one in which the potential patch was completely masked.

**Theorem 2.** *Assume that $v(x)_{i,j} = 1$. Then*

$$\forall \, (p, \, l) \in \mathcal{P} : v(A(x, p, l))_{i,j} = 1 \Rightarrow f(A(x, \, p, \, l))_{i,j} = f(x)_{i,j}$$

See the proofs for both theorems in Appendix A. For a given image $x$ the verification map $v(x)$ is complementary to the model segmentation output $f(x)$ that stays unchanged. Thus, there is no drop in clean performance however we may have some false positive alerts in the clean setting.

## 5 Experiments

In this section, we evaluate DEMASKED SMOOTHING with the masking schemes proposed in Section 4, compare our approach with the direct application of Derandomized Smoothing [9] to the segmentation task and evaluate the performance on different datasets and models. Certified recovery and certified detection provide certificates of different strength (Section 4) which are not comparable. We evaluate them separately for different patch sizes.

**Experimental Setup**. We evaluate DEMASKED SMOOTHING on two challenging semantic segmentation datasets: ADE20K [31] (150 classes, 2000 validation images) and COCO-Stuff-10K [39] (171 classes, 1000 validation images). For demasking we use the ZITS [29] inpainting model with the checkpoint trained on Places2 [40] from the official paper repository [1]. As a segmentation model $f$ we use BEiT [30], Swin [26], PSPNet [24] and DeepLab v3 [41]. We use the model implementations provided in the *mmsegmentation* framework [42]. An illustration of the image reconstruction and respective segmentation can be found in Figure 3.

**Evaluation**. We compute mIoU, mean recall (mR) and certified mean recall (cmR). See detailed explanation of these metrics in Appendix C. In certified detection, we additionally consider false alert ratio (FAR) which is the fraction of correctly classified pixels for which we return an alert on a clean image. Smaller FAR is preferable. Due to our threat model, certifying small objects in the scene can be difficult because they can be partially or completely covered by an adversarial patch. To provide an additional perspective on our methods, we also evaluate mR and cmR specifically for the "big" classes, which occupy on average more than 20% of the images in which they appear. These are, for example, road, building, train, and sky, which are important for understanding the scene. The full list is provided in the Appendix I. See execution time in Table 11 in the Appendix J.

**Discussion**. In Table 1, we compare different masking schemes proposed in Section 4.1. Evaluation of all the models with all the masking schemes is consistent with these results and can be found in Appendix F. We see that column masking achieves better results in both certification modes. We

---

[1] https://github.com/DQiaole/ZITS_inpainting

Table 1: Comparison of different masking schemes proposed in Section 4.1. mIoU - mean intersection over union, mR - mean recall, cmR - certified mean recall. %C - mean percentage of certified and correct pixels in the image. For detection, we provide clean mIoU since the output is unaffected and mean false alert rate (FAR) (lower is better). See additional results in Appendix F.

| dataset | segm | mode | mask | mIoU | big mR | big cmR | all mR | all cmR | %C | FAR ↓ |
|---|---|---|---|---|---|---|---|---|---|---|
| ADE20K | BEiT-B | detection 1% patch | column | 53.08 | 70.92 | **57.33** | 64.45 | **32.55** | **63.55** | **20.04** |
| | | | row | | | 50.05 | | 26.65 | 58.34 | 25.24 |
| | | recovery 0.5% patch | column | **24.92** | **60.77** | **41.26** | **29.84** | **12.98** | **46.22** | N/A |
| | | | row | 16.33 | 46.91 | 16.72 | 19.51 | 4.83 | 31.71 | |
| | | | 3-mask | 19.90 | 56.90 | 26.51 | 23.86 | 7.54 | 38.64 | |
| | | | 4-mask | 18.82 | 52.96 | 23.75 | 22.56 | 5.87 | 34.36 | |

Table 2: Demasked Smoothing results with column masking for different models

| mode | dataset | segm | mIoU | big mR | big cmR | all mR | all cmR | %C | FAR ↓ |
|---|---|---|---|---|---|---|---|---|---|
| detection 1 % patch | ADE20K | BEiT-B | **53.08** | **70.92** | **57.33** | **64.45** | **32.55** | **63.55** | 20.04 |
| | | PSPNet | 44.39 | 61.83 | 50.02 | 54.74 | 26.37 | 60.57 | 20.08 |
| | | Swin-B | 48.13 | 68.51 | 55.45 | 59.13 | 29.06 | 61.44 | 20.31 |
| | COCO10K | PSPNet | 37.76 | 71.71 | 56.86 | 49.65 | 26.80 | 47.09 | 21.43 |
| | | DeepLab v3 | 37.81 | 72.52 | 56.54 | 49.98 | 26.86 | 46.55 | 21.89 |
| recovery 0.5 % patch | ADE20K | BEiT-B | **24.92** | **60.77** | **41.26** | **29.84** | **12.98** | **46.22** | N/A |
| | | PSPNet | 19.17 | 51.90 | 34.11 | 23.66 | 10.76 | 44.90 | |
| | | Swin-B | 22.43 | 59.75 | 34.88 | 27.09 | 11.70 | 46.14 | |
| | COCO10K | PSPNet | 21.94 | 61.56 | 36.67 | 29.94 | 11.13 | 29.51 | |
| | | DeepLab v3 | 23.12 | 62.60 | 33.84 | 31.59 | 11.55 | 28.71 | |

attribute the effectiveness of column masking to the fact most of the images in the datasets have a clear horizon line, therefore having a visible column provides a slice of the image that intersects most of the scene background objects.

In Table 2, we evaluate our method with column masking on different models. For certified detection we can certify more than 60% of the pixels with all models on ADE20K and more than 46 % on COCO10K. False alert ratio on correctly classified pixels is around 20%. In certified recovery, we certify more than 44% pixels on ADE20K and more than 28% pixels on COCO10K. See the comparison with DRS [9] adaptded for segmentation in Appendix H. We evaluate the performance of our method for different patch sizes in Section C. Ablations with respect to inpainting can be found in Appendix G. DEMASKEDSMOOTHING illustrations procedure are provided in Appendix L.

## 6 Conclusion

In this work, we propose DEMASKED SMOOTHING, the first (up to our knowledge) certified defence framework against patch attacks on segmentation models. Due to its novel design based on masking schemes and image demasking, DEMASKED SMOOTHING is compatible with any segmentation model and can on average certify 63% of the pixel predictions for a 1% patch in the detection task and 46% against a 0.5% patch for the recovery task on the ADE20K dataset.

**Ethical and Societal Impact** This work contributes to the field of certified defences against physically-realizable adversarial attacks. The proposed approach allows to certify robustness of safety-critical applications such as medical imaging or autonomous driving. The defence might be used to improve robustness of systems used for malicious purposes such as (semi-)autonomous weaponry or unauthorized surveillance. This danger may be mitigated e.g. by using a system of sparsely distributed patches which makes certifying the image more challenging. All activities in our organization are carbon neutral, so our experiments do not leave any carbon dioxide footprint.

## Acknowledgements

We thank Chong Xiang for the suggestions on extending our evaluation protocol. Matthias Hein is a member of the Machine Learning Cluster of Excellence, EXC number 2064/1 – Project number 390727645 and of the BMBF Tübingen AI Center, FKZ: 01IS18039B.

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

## A Proofs (Section 4)

In this section, we provide the proofs for the theorems stated in Section 4.

**Certified recovery.** For the threat model $\mathcal{P}$ (Section 3) consider a set $M$ of $K$ masks. We define a function $h : \mathcal{X} \to \mathbb{S}$ that assigns a class to the pixel $x_{i,j}$ via majority voting over class predictions of each reconstructed segmentation in $S$. A class for the pixel that is predicted by the largest number of segmentations is assigned. We break the ties by assigning a class with a smaller index.

**Theorem 1**. (Section 4.2) If the number of masks $K$ satisfies $K \geq 2T(M) + 1$ and for a pixel $x_{i,j}$ we have

$$\forall\, S[k] \in S : \ S[k]_{i,j} = h(x)_{i,j}$$

i.e. all the votes agree, then $\forall\, (p,\, l) \in \mathcal{P} : \ h(A(x, p, l))_{i,j} = h(x)_{i,j}$.

*Proof.* Assume that

$$\exists\, (p,\, l) \in \mathcal{P} : \ h(A(x, p, l))_{i,j} \neq h(x)_{i,j}$$

Let us denote $x' := A(x, p, l)$ and $S'$ to be the segmentation array for $x'$. Then the class $h(x)_{i,j}$ did not get the majority vote for $S'$. However, by definition of $T(M)$ we know that $(p,\, l)$ could affect at most $T(M)$ segmentations. Since all $K$ segmentations of $S$ have voted for $h(x)_{i,j}$, then at least $K - T > \frac{K}{2}$ of them are still voting for $h(x)_{i,j}$ in $S'$ meaning that $h(x)_{i,j}$ still has the majority vote in $S'$. Therefore $h(x')_{i,j} = h(x)_{i,j}$ ☐

**Certified detection.** Consider $M_d = \{M_d[k]\}_{k=1}^K$. For a set of demasked segmentations S we define the verification map $v(x)_{i,j} := [f(x)_{i,j} = S[1]_{i,j} = \ldots = S[K]_{i,j}]$ i.e. the original segmentation coincides with all the other segmentations including the one in which the potential patch was completely masked.

**Theorem 2**. (Section 4.2) Assume that $v(x)_{i,j} = 1$. Then

$$\forall\, (p,\, l) \in \mathcal{P} : v(A(x, p, l))_{i,j} = 1 \Rightarrow f(A(x,\, p,\, l))_{i,j} = f(x)_{i,j}$$

*Proof.* Assume that $\exists\, (p,\, l) \in \mathcal{P}$ s. t. $v(A(x, p, l))_{i,j} = 1$ and $f(A(x,\, p,\, l))_{i,j} \neq f(x)_{i,j}$. Let us denote $x' := A(x, p, l)$ and $S'$ to be the segmentation set for $x'$. By definition of $M_d$, $\exists\, M_d[k] \in M_d$ s. t. $M_d[k]$ masks the patch $(p,\, l)$ Hence,

$$g(x \odot M_d[k]) = g(x' \odot M_d[k]),$$

$$S[k] = f(g(x \odot M_d[k])) = f(g(x' \odot M_d[k])) = S'[k],$$

Since $v(x)_{i,j} = 1$, we have $f(x)_{i,j} = S[k]_{i,j}$. Since $v(x')_{i,j} = 1$, we have $f(x')_{i,j} = S'[k]_{i,j}$. Thus, $f(x')_{i,j} = f(x)_{i,j}$. ☐

## B Detailed description of masking strategies

In this section, we provide additional details for constructing certified recovery masks proposed in Section 4.1.

**Certified recovery**. We define mask sets $M$ that satisfy different values of $T$. We divide the image $x$ into a set of non-intersecting blocks $B$ of the same size as an adversarial patch, $H' \times W'$ (see Figure 5), $1 \leq q \leq \lceil H/H' \rceil$, $1 \leq r \leq \lceil W/W' \rceil$. In each mask, each of these blocks will be either masked or not masked (i. e. *visible*). Moreover, for each block there exists only one mask in which it is visible. For a set $M$ of $K$ masks we define the mapping $\mu_M : B \to \{1, \ldots, K\}$. If $\mu(B[q,\, r]) = k$, then $B[q,\, r]$ is not masked in $M[k]$. Therefore, each mask $M[k]$ is defined by a $B_k \subset B$ s. t. for $b \in B_k$ $\mu(b) = k$.

We define a set $M$ that we call 3-mask for which $T(M) = 3$. We assign the blocks in each row to the masks as follows: $\mu(B[1,\, 1]) = 1$; $\mu(B[1,\, 2]) = \mu(B[1,\, 3]) = 2$; $\mu(B[1,\, 4]) = \mu(B[1,\, 5]) = 3$ and so on until we reach the end of the row. If we finish the first row with the value $k$, then we start the second row as follows $\mu(B[2,\, 1]) = \mu(B[2,\, 2]) = k + 1$; $\mu(B[2,\, 3]) = \mu(B[2,\, 4]) = k + 2$: …. If we finish the second row on $n$, we start the third row similarly to the first: $\mu(B[3,\, 1]) = n + 1$; $\mu(B[3,\, 2]) = \mu(B[3,\, 3]) = n + 2$; … When we reach the number $K$, we start from 1 again

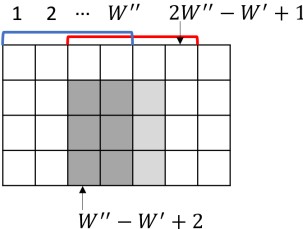

Figure 4: The masked columns of the first two adjacent masks (blue for the first one and red for the second one). If the patch is not completely masked by the first mask, it should be visible at the column $W'' + 1$ (the masked part of the patch is dark-grey and the visible part is in light-grey). However then the patch will be completely masked by the second mask.

(Figure 5d). Due to the block size, the patch cannot intersect more than four blocks at once. Our parity-alternating block sequence ensures that in any such intersection of four blocks either the top ones or the bottom ones will belong to the same masking, so at most three different maskings can be affected.

We define a set $M$ that we call 4-mask for which $T(M) = 4$. Due to our block size any assignment of masks will work because the patch cannot intersect more than four blocks. We consider the one that allows uniform distribution of the unmasked blocks (Figure 5g). We point out that for the described methods each masking keeps approximately $1/K$ of the pixels visible and the unmasked regions are uniformly distributed in the image. This means that for any masked pixel there exists an unmasked region located close enough to this pixel. It is the core difference between our masks and the ones proposed for certified classification such as block or column smoothing [9]. It was observed that the image demasking is facilitated when the visible regions are uniformly spread in the masked image [36]. We present the full demasked smoothing procedure in Algorithm 1.

**Certified detection**. We define $M_d$ to be a set of masks for certified detection (we use subscript $d$ for distinction). $M_d$ should have the property: $\forall \, (p, l) \in \mathcal{P} \, \exists \, m \in M_d : A(x, \, p, \, l) \odot m = x \odot m$ i. e. for every patch exists at least one mask not affected by this patch. See deteils in Appendix B. For a patch of size $H' \times W'$ we consider $K = W - W' + 1$ masks such that the mask $M_d[k]$ masks a column of width $W'$ starting at the horizontal position $k$ in the image (Figure 2e). To obtain the guarantee for the same $\mathcal{P}$ with a smaller $K$, we consider a set of strided columns of width $W'' \geq W'$ and stride $W'' - W' + 1$ that also satisfy the condition.

**Lemma 1.** *Consider an image of the size $H \times W$. Let $H' \times W'$ be a fixed adversarial patch size. Let $M^d(K, \mathcal{L})$ be a set of masks where each mask is masking an $H \times W''$ vertical column, $W'' \geq W'$. Let the stride between the columns in two adjacent masks be $W'' - W' + 1$. Then for any location $l \in \mathcal{L}$ of the patch, there exists a mask that covers it completely.*

*Proof.* (Adapted from the proof of Lemma 4 in PatchCleanser [14]). Without loss of generality, we consider the first two adjacent column masks. The first one covers the columns from 1 to $W''$. The second mask covers the columns from $1 + (W'' - W' + 1) = W'' - W' + 2$ to $(W'' - W' + 2) + (W'' - 1) = 2W'' - W' + 1$ (See Figure 4). Now consider an adversarial patch of size $H' \times W'$. Let us find the smallest possible start index of this patch so that it does not get covered by the first mask. For that it should be visible at the column $W'' + 1$ and, therefore, start at the column with index not smaller than $(W'' + 1) - W' + 1 = W'' - W' + 2$. However, it is the same column in which second mask starts. Therefore, given that $W'' \geq W'$ we have that the patch is completely masked by the second mask. Then for a patch which is only partially masked by the second mask from the left we use an analogous argument to show that it is completely masked by the third mask and so on. $\qquad\square$

A similar scheme can be proposed for the rows (Figure 2f). Alternatively, we could use a set of block masks of size $H' \times W'$. Then the number of masks grows quadratically with the image resolution. Hence, in the experiments we focus on the column and the row masking schemes.

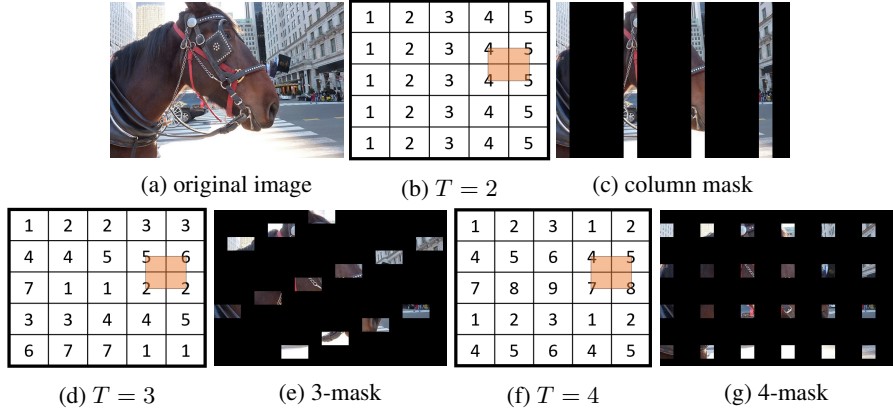

(a) original image      (b) $T = 2$      (c) column mask

(d) $T = 3$      (e) 3-mask      (f) $T = 4$      (g) 4-mask

Figure 5: (a) examples of a mask for the column masks with $T = 2$ (b, c), 3-mask with $T = 3$ (d, e), and 4-mask with $T = 4$ (f, g) with the number of masks $K = 5, 7, 9$ respectively. The number of a block denotes in which mask it is not masked (there is only one such mask for each block). For each mask set, we show one of the locations $l$ in which an adversarial patch affects $T$ different maskings.

---

**Algorithm 1** Demasked Smoothing

---

**Input:** image $x \in [0, 1]^{H \times W \times C}$, patch size $(H', W')$, certification type CT (recovery or detection), mask type MT (column, row, 3-mask, 4-mask), inpainting model $g$, segmentation model $f$
**Output:** segmentation map $h \in \mathcal{Y}^{H \times W}$, certification (or verification) map $v \in \{0, 1\}^{H \times W}$

1:   $M \leftarrow \text{CreateMaskArray}(H, W, H', W', \text{CT}, \text{MT})$           ▷ according to section 4.1
2: **for** $k \leftarrow 1, \ldots, |M|$ **do**                    ▷ this loop can be paralellized
3:      $S[k] \leftarrow f(g(x \odot M[k]))$ ▷ mask input, inpaint the masked regions, and apply segmentation
4: **end for**
5: **if** CT = 'recovery' **then** $h \leftarrow \text{MajorityVote}(S)$   ▷ vote over the classes predicted for each pixel
6: **else**    $h \leftarrow f(x)$                    ▷ in detection case, output clean segmentation
7: **end if**
8: $v \leftarrow \text{AllEqual}(S, h)$      ▷ assign 1 for the pixels where all $S[k]$ agree with $h$ and 0 otherwise
9: **Return** $h, v$

---

## C   Evaluation metrics

For both certified recovery and certified detection, we provide a standard segmentation output (without any abstention) and a corresponding certification map (Figure 3). In case of certified detection, the segmentation output remains the same as for the original segmentation model, however, there may be false alerts in the certification map. For the certified recovery, the output is obtained by a majority vote over the segmentations of demasked images (Section 4.2). We evaluate the mean intersection over union (mIoU) for these outputs. The certification map is obtained by assigning to each certified pixel the corresponding class from the segmentation output and assigning a special *uncertified* label to all non-certified pixels. For each image we evaluate the fraction of pixels which are certified and correct (coincide with the ground truth). %C is a mean of these fractions over all the images in the dataset. In semantic segmentation task, the class frequencies are usually skewed, therefore global pixel-wise accuracy alone is an insufficient metric.

Matching the certification map separately for each class $y \in \mathcal{Y}$ with the ground truth segmentation for $y$ in the image $x$ allows us to compute the guaranteed lower bound ($cTP_y(x)$) on the number of true positive pixel predictions ($TP_y(x)$) i.e. those that were correctly classified into $y$. If a pixel was certified with a correct class, then this prediction cannot be changed by a patch (or, alternatively, the change will be detected by the verification function $v$ in certified detection). We consider *recall* $R_y(x) = \frac{TP_y(x)}{TP_y(x) + FN_y(x)}$ where $FN_y(x)$ is the number of false negative predictions for $y$ in $x$. $P_y(x) = TP_y(x) + FN_y(x)$ is the total area of $y$ in the ground truth and does not depend on our prediction. We can evaluate certified recall $cR_y(x) = \frac{cTP_y(x)}{P_y(x)}$, a lower bound on the recall $R_y(x)$. Total recall and certified total recall of class $y$ in a dataset $D$ are $TR_y(D) =$

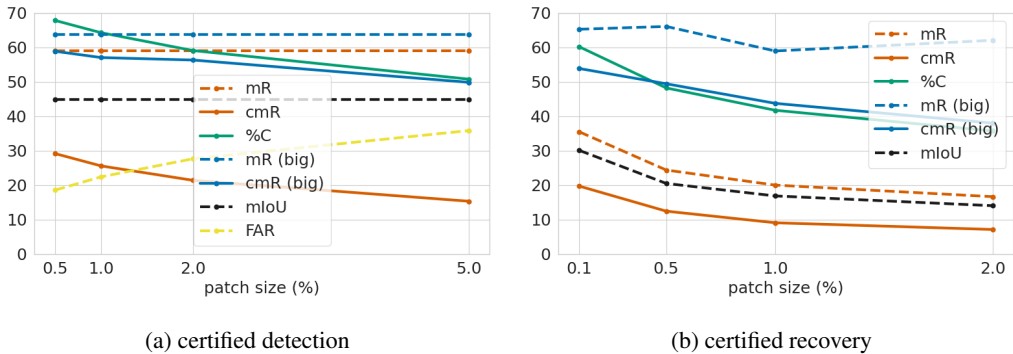

(a) certified detection

(b) certified recovery

Figure 6: Performance for different adversarial patch sizes evaluated on 200 ADE20K images.

$\frac{\sum_{x \in D} TP_y(x)}{\sum_{x \in D} P_y(x)}$ and $cTR_y(D) = \frac{\sum_{x \in D} cTP_y(x)}{\sum_{x \in D} P_y(x)}$ respectively. Then, we obtain mean recall $mR(D) = \frac{1}{|\mathcal{Y}|} \sum_{y \in \mathcal{Y}} TR_y(D)$ and certified mean recall $cmR(D) = \frac{1}{|\mathcal{Y}|} \sum_{y \in \mathcal{Y}} cTR_y(D)$. Evaluating lower bounds on other popular metrics such as mean precision or mIoU this way results in vacuous upper bound since they depend on the upper bound on false positive ($FP$) predictions. For the pixels that are not certified we cannot guarantee that they will not be assigned to a certain class, therefore, a non-trivial upper bound on $FP$ is not straightforward. We leave this direction for future work. In certified detection, we additionally consider false alert ratio (FAR) which is the fraction of correctly classified pixels for which we return an alert on a clean image. Smaller FAR is preferable.

Figure 6 shows how the performance of DEMASKED SMOOTHING depends on the patch size for the BEiT-B model. We see that certified detection metrics remain high even for a patch as big as 5% of the image surface and for the recovery they slowly deteriorate as we increase the patch size to 2%.

## D    Test-time input certification

In this section, we discuss how certified recovery (Theorem 1) can be applied to guaranteed verification of the robustness on a test image. We also discuss how robustness guarantees for the test-time images can be evaluated by using a dataset of clean images such as ADE20K [31] or COCO-Stuff-10K [39].

### D.1    Test-time certified recovery

Let $x'$ be a test-time input which can be either a clean image or an image attacked with an adversarial patch. We know that there exists a clean image $x$ corresponding to $x'$ which removes the patch if it is present. We have either $x' = x$ or $x' \in A(x)$, where $A(x) := \{A(x, p, l) \mid (p, l) \in \mathcal{P}\}$. However, at test time we do not have access to the clean image $x$.

Our goal is to certify that for our segmentation model $h$ and a pixel $x_{i,j}$ we have $h(x')_{i,j} = h(x)_{i,j}$. We can achieve this result by applying the recovery certification (Theorem 1) to the test-time image. It allows us to verify whether $\forall (p, l) \in \mathcal{P} : h(A(x', p, l))_{i,j} = h(x')_{i,j}$. We also know that if $x' \in A(x)$, then $x \in A(x')$ (Figure 7a). Indeed, if $x'$ is only different from $x$ by one patch, then $x$ can be be obtained from $x'$ by removing this patch. Therefore, by obtaining the guarantee for $A(x')$, we implicitly obtain the guarantee also for the image $x$ even though we do not have direct access to it.

We note that this test-time guarantee is only possible for certified recovery. In certified detection, we would need to evaluate the verification function $v$ (Theorem 2) for both the clean image $x$ and the attacked image $x'$ to obtain the result. This cannot be done if $x$ is implicit.

### D.2    Robustness guarantees evaluation

The typical certified robust error for a given test data set (and pixel $(i, j)$ in the segmentation case) is an estimate for

$$\mathbb{E}_{X \sim D} \big[ \max_{(p,l) \in P} \mathbb{1}_{h(A(X,p,l))_{i,j} \neq h(X)_{i,j}} \big],$$

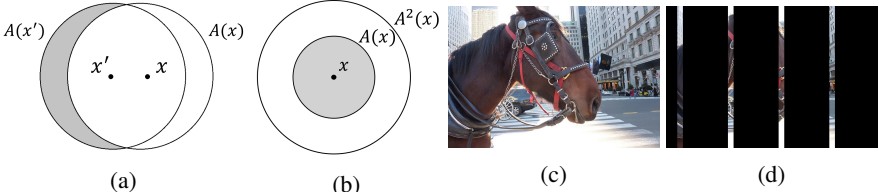

Figure 7: (a) certified inference; (b) double adversarial neighbourhood; (c) original image (d) certification against two patches

where $D$ is the data generating probability measure and we assume that our test set to be an i.i.d. sample of it. This is the expected robust error (worst case over our threat model $P$ for clean inputs) for a given pixel $(i, j)$. Using the test sample to get an estimate of this quantity, we get a probabilistic guarantee that the corresponding pixel $(i, j)$ of a new *clean* test sample $x'$ drawn i.i.d. from $P$ will have its whole "patch"-neighborhood certified.

However, more important for a practical security analysis is that we can certify a given instance, which can be even potentially adversarially perturbed. Formally, this means that for an input $z \in A(x)$, where $x \sim P$ is an unknown sample from $P$, that we guarantee

$$\forall (p, l) \in P : h(A(z, p, l))_{i,j} = h(z)_{i,j},$$

and as $x \in A(z, p, l)$ this implies that we certify that the pixel $(i, j)$ of the potentially manipulated image is classified the same as pixel $(i, j)$ of the unperturbed image $x$.

However, it is now tricky to get even a probabilistic estimate of the quantity

$$\mathbb{E}_{x \sim D} \max_{(p,l) \in P} \left[ \max_{(q,m) \in P} \mathbb{1}_{h(A(A(x,p,l),q,m))_{i,j} = h(A(x,p,l))_{i,j}} \right],$$

as the outer maximization process cannot be simply simulated by doing adversarial patch attacks on a clean test dataset.

We propose a way to evaluate a guaranteed lower bound on the fraction of certified test-time inputs by using a dataset of clean images. Instead of considering a standard one-patch neighbourhood $A(x)$ defined by our threat model (Section 3), we propose to consider a neighbourhood $A^2(x)$ of two independent patches (Figure 7b). $A^2(x)$ contains all the images $x' \in A(x)$ as well as their respective patch neighbourhoods $A(x')$. Therefore, by verifying that $\forall (p_1, l_1), (p_2, l_2) \in \mathcal{P} : h(A(A(x, p_1, l_1), p_2, l_2))_{i,j} = h(x)_{i,j}$, we guarantee that $\forall x' \in A(x) \forall (p, l) \in \mathcal{P} : h(A(x', p, l))_{i,j} = h(x')_{i,j}$.

We note that corresponding reasoning could be applied to certification in $\ell_p$ models. Then $A^2(x)$ would correspond to doubling the radius of the $\epsilon$-ball instead of adding a second patch.

Note that Theorem 1 can be directly extended to a threat model of $N$ patches. In the worst case each of the $N$ patches can affect $T$ different maskings. Therefore, we need to change the condition of Theorem 1 to $K \geq 2NT + 1$. We apply the described method to evaluating the test-time certification guarantees for a toy example of a $0.1\%$ patch in Table 3. We also illustrate how a column mask looks in this case in Figure 7.

Table 3: Inference recovery robustness estimate. To illustrate our point we certify an example for a $0.1\%$ patch.

| dataset | segm | mask | mIoU | big | | all | | %C |
|---------|------|------|------|-----|-----|-----|-----|-----|
| | | | | mR | cmR | mR | cmR | |
| ADE20K | BEiT-B | col | 19.73 | 36.95 | 16.64 | 24.23 | 9.24 | 41.96 |
| COCO10K | | | 26.36 | 69.63 | 35.34 | 34.92 | 11.13 | 28.17 |

# E  Adversarial patch example

In this section, we demonstrate an example of a real adversarial patch for a semantic segmentation model similar to the one illustrated in the Figure 1a and show how it is handled by our certified

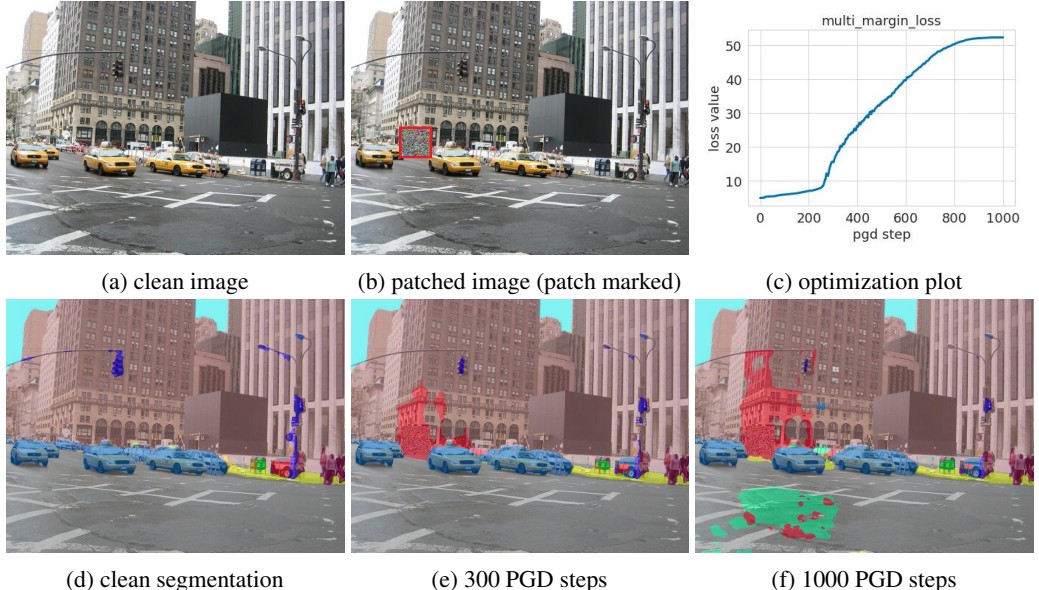

Figure 8: Patch attack illustration with Swin [26] and an ADE20K image. A patch occupying 1% of the image surface changes the segmentation.

defences. We illustrate it for the Swin [26] model on one of the images from the ADE20K [31] dataset.

### E.1 Patch optimization

We set the patch size to 1% of the image surface. We select a fixed position for a patch on the rear window of a car (Figure 8a). For each pixel we extract a list of predicted logits corresponding to each class and apply multi-margin loss with respect to the ground truth label of the respective pixel. We use random patch initialization without restarts. As an optimizer we use projected gradient descent (PGD) with 1000 steps and initial step size of 0.01. We use cosine step size schedule and momentum for the gradient with the rate of 0.9. The optimization plot and the patch efficiency at different iterations of the PGD are illustrated in the Figure 8.

### E.2 Certified recovery

We denote the original image as $x$ and the patched image as $x'$. The voting-based segmentation function $h$ (Section 4.2) provides the majority-vote prediction $h(x)$ and the corresponding certification map which shows the pixels where all the votes agree. In Figure we see that a part of the building and the road is certified which means that this prediction cannot be affected by an adversarial patch. Figure demonstrates $h(x')$ which correctly segments those regions in presence of an adversarial patch that fools the original model.

### E.3 Certified detection

We perform our analysis by evaluating the verification map $v$ (Section 4.2) for the original image $x$ and for the patched image $x'$. We see that in $v(x)$ a major part of the building is certified i. e. for a part of pixels $x_{i,j}$ that belong to the building and the road we have $v(x)_{i,j} = 1$. However, $v(x')_{i,j} = 0$ for those pixels. It means that we have detected that the prediction on this input is potentially affected by an adversarial patch.

## F Additional experiments

In Tables 4 and 5, we provide additional experimental results for evaluating different masking schemes proposed in Section 4.1 on different models.

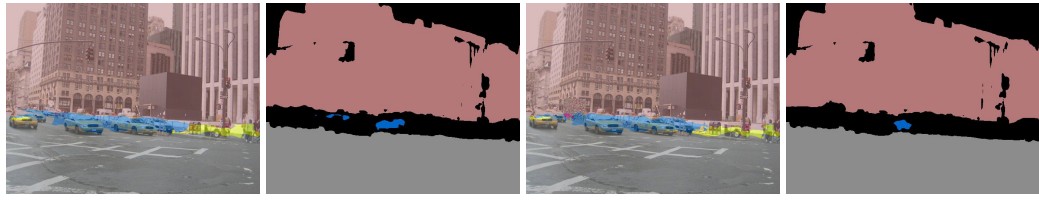

(a) segmentation of the original image, $h(x)$
(b) certification map of the original image $x$
(c) segmentation of the patched image, $h(x')$
(d) certification map of the patched image $x'$

Figure 9: Certified recovery for a 1 % patch used in the attack. The majority vote function $h$ recovers the prediction in presence of an adversarial patch that fools the undefended model. The segmentation for the original and patched image in (a) and (c) are the same for the regions certified in the certification maps (b) or (d). The certification maps (b) and (d) are also almost the same.

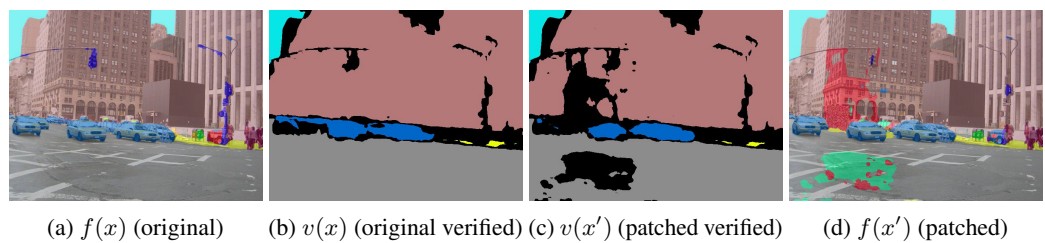

(a) $f(x)$ (original)  (b) $v(x)$ (original verified) (c) $v(x')$ (patched verified)  (d) $f(x')$ (patched)

Figure 10: $f$ is a segmentation model (Swin [26]) and $v$ is the verification function (Section 4.2). For an attacked image $x'$ $v(x')$ detects the region of $f(x')$ which was (potentially) affected by an adversarial patch.

## G  Inpainting ablation studies

We perform ablation studies with respect to the demasking step. The results are in Table 6. Figure 11 provides additional illustrations. As can be seen from the results, our method heavily benefits from having available stronger inpainting models that allow achieving better clean and certified accuracy. We consider this property actually as a strength of our method since it will automatically benefit from future research and developments of stronger inpainting methods. For certified recovery, we also compare it to GIN [43] based on a generative model that we trained on ADE20K (without using style losses based on ImageNet trained VGG). The results are in Table 7. Illustrations can be found in Figure 12.

Table 4: The certified detection results(%) for a patch occupying no more than 1% of the image. mIoU - mean intersection over union, mR - mean recall, cmR - certified mean recall. %C - mean percentage of certified and correct pixels in the image.

| dataset | segm | mask | mIoU | big mR | big cmR | all mR | all cmR | %C |
|---|---|---|---|---|---|---|---|---|
| ADE20K | PSPNet | col | 44.39 | 61.83 | **50.02** | 54.74 | **26.37** | **60.57** |
|  |  | row |  |  | 42.44 |  | 19.88 | 54.62 |
|  | Swin | col | 48.13 | 68.51 | **55.45** | 59.13 | **29.06** | **61.44** |
|  |  | row |  |  | 47.21 |  | 22.04 | 55.93 |
| COCO10K | PSPNet | col | 37.76 | 71.71 | **56.86** | 49.65 | **26.80** | **47.61** |
|  |  | row |  |  | 51.05 |  | 23.51 | 43.40 |
|  | DeepLab v3 | col | 37.81 | 72.52 | **56.54** | 49.98 | **26.86** | **47.17** |
|  |  | row |  |  | 50.51 |  | 23.89 | 43.19 |

Table 5: The certified recovery results(%) against a 0.5% patch. 3-mask and 4-mask correspond to $T = 3$ and $T = 4$ respectively (Figure 2)

| dataset | segm | mask | mIoU | big | | all | | %C |
|---|---|---|---|---|---|---|---|---|
| | | | | mR | cmR | mR | cmR | |
| ADE20K | PSPNet | col | **19.17** | **51.90** | **34.11** | **23.66** | **10.76** | **44.90** |
| | | row | 12.00 | 36.26 | 12.03 | 15.03 | 3.74 | 28.29 |
| | | 3-mask | 15.00 | 44.93 | 19.55 | 18.41 | 5.58 | 35.85 |
| | | 4-mask | 12.74 | 40.41 | 15.86 | 15.87 | 4.14 | 31.22 |
| | Swin | col | **22.43** | **59.75** | **34.88** | **27.09** | **11.70** | **46.14** |
| | | row | 13.58 | 42.88 | 15.13 | 16.70 | 4.46 | 30.64 |
| | | 3-mask | 17.06 | 51.03 | 24.15 | 20.74 | 6.65 | 38.27 |
| | | 4-mask | 14.77 | 46.67 | 17.74 | 10.05 | 4.72 | 34.04 |
| COCO10K | PSPNet | col | **21.94** | **61.56** | **36.67** | **29.94** | **11.13** | **29.51** |
| | | row | 18.87 | 58.04 | 20.90 | 26.16 | 6.14 | 19.31 |
| | | 3-mask | 18.82 | 59.26 | 29.00 | 25.85 | 7.56 | 25.21 |
| | | 4-mask | 17.46 | 58.47 | 23.63 | 24.35 | 5.51 | 20.36 |
| | DeepLab v3 | col | **23.12** | **62.60** | **33.84** | **31.59** | **11.55** | **28.71** |
| | | row | 20.04 | 55.71 | 17.80 | 27.89 | 6.28 | 17.04 |
| | | 3-mask | 20.14 | 58.02 | 27.14 | 27.82 | 8.05 | 24.30 |
| | | 4-mask | 19.35 | 58.22 | 22.01 | 26.74 | 5.79 | 19.38 |

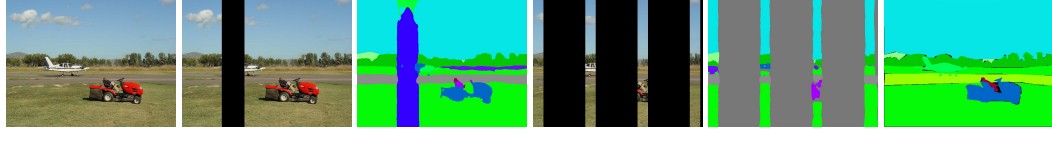

(a) original    (b) detection col  (c) segmentation  (d) recovery col  (e) segmentation  (f) ground truth

Figure 11: Results without image demasking. The solid color inpainting is treated as a separate object in the scene because we need to classify every pixel in semantic segmentation task. Therefore, it is hard to achieve a situation where all the demasked segmentation agree on some pixel which is represented in the Table 6.

Table 6: Comparison for demasked smoothing with and without demasking step. mIoU - mean intersection over union, mR - mean recall, cmR - certified mean recall. %C - mean percentage of certified and correct pixels in the image. We use Swin model on 200 ADE20K images with column masking for certified detection and certified recovery. We compare masking the columns with solid black color without demasking to ZITS demasking.

| mode | patch size | demasking | mIoU | big | | all | | %C |
|---|---|---|---|---|---|---|---|---|
| | | | | mR | cmR | mR | cmR | |
| detection | 1.0% | ✓ | 38.56 | 67.25 | **58.85** | 53.37 | **23.35** | **62.89** |
| | | ✗ | | | 19.49 | | 3.09 | 21.19 |
| recovery | 0.5% | ✓ | **19.09** | **66.03** | **52.71** | **23.02** | **12.66** | **47.05** |
| | | ✗ | 1.10 | 15.09 | 7.71 | 1.79 | 0.72 | 18.59 |

Table 7: Comparison of our two demasking methods: ZITS and GIN. mIoU - mean intersection over union, mR - mean recall, cmR - certified mean recall. %C - mean percentage of certified and correct pixels in the image. We use Swin model on 200 ADE20K images.

| demasking | trained on | mIoU | big | | all | | %C |
|---|---|---|---|---|---|---|---|
| | | | mR | cmR | mR | cmR | |
| ZITS [29] | Places2 | **19.09** | **66.03** | **52.71** | **23.02** | **12.66** | **47.05** |
| GIN [43] | ADE20K | 5.46 | 32.27 | 19.05 | 7.62 | 3.52 | 32.08 |

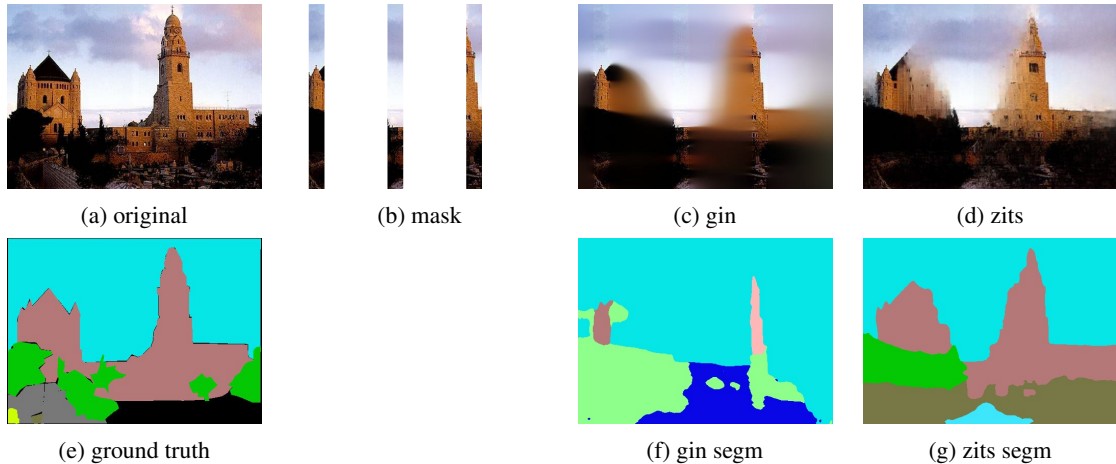

(a) original        (b) mask        (c) gin        (d) zits

(e) ground truth        (f) gin segm        (g) zits segm

Figure 12: Comparison between GIN and ZITS inpainting

## H    Comparison to simplified Derandomized Smoothing

Derandomized Smoothing (DRS) [9] was proposed for certified recovery, therefore in this section we focus on this task. Direct adaptation of derandomized smoothing to semantic segmentation task requires training a model that is able to predict the full image segmentation from a small visible region. Since it is not immediately clear to us what architectural design and training procedure would be needed to train such a model, we consider a simplified version of DRS that we call DRS-S. In this version, we consider an off-the-shelf semantic segmentation model and evaluate how it performs with column masking from DRS. Therefore, we do not encode the masked regions with the special 'NULL' value like in DRS but use black color instead. That is because an off-the-shelf model cannot work with 'NULL' values.

We run our experiments on ADE20K dataset. We consider the DRS parameters from the recent SOTA version of Derandomized Smoothing by Salman et al. [15]. They use column width $b = 19$ and stride $s = 10$ for certified classification of 224x224 ImageNet images. To account for the fact that ADE20K images have larger resolution than ImageNet, we scale the parameters to column width $b = 42$ and stride $s = 22$. To make the comparison consistent with the rest of our results, we use the patch occupying $0.5\%$ of the image.

From Table 8 we can see that DRS-S performs poorly on semantic segmentation task. The reason for that is illustrated in Figure 13. Processing the column region in 13c would probably be sufficient for a classification model to classify the image into the class "house". But it is clearly not sufficient to reconstruct the whole segmentation map 13e as can be seen in the Figure 13g. Whether doing this would be possible with a model specifically trained to reconstruct the segmentation map from a very small visible region is an open research question (up to our knowledge).

We point out that the value %C of certified and correctly classified pixels in the Table 8 is still surprisingly high for DRS-S compared to other metircs. We attribute this to the fact that the solid black regions are usually treated as a wall by the segmentation model, therefore the images are usually segmented as a wall by the DRS majority voting. And the wall is a common part of both indoor and outdoor scenes in ADE20K as can be implied from the Table 9 of "big" ADE20K classes. Therefore, always classifying the output as a wall provides a decent fraction of correctly classified pixels because of the skewed classes.

However, to provide a better comparison with DRS, we emulate the model which is able to reconstruct the whole segmentation map from the column masking proposed in DRS. We do this by applying the demasking approach proposed in this work. We first try to reconstruct the whole image from one column and then segment it with an off-the-shelf model as we did with the masks proposed in this paper. We call this approach DRS-E and the results can be found in Table 8.

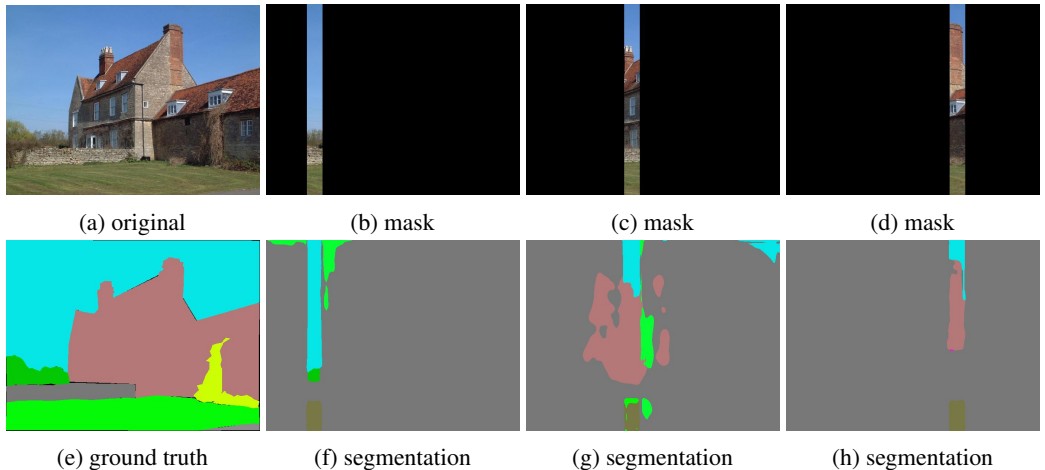

|        |        |        |        |        |
| :----: | :----: | :----: | :----: | :----: |
| (a) original | (b) mask | (c) mask | (d) mask |
| (e) ground truth | (f) segmentation | (g) segmentation | (h) segmentation |

Figure 13

Table 8: Comparison of our method with simplified Derandomized Smoothing [9]. mIoU - mean intersection over union, mR - mean recall, cmR - certified mean recall. %C - mean percentage of certified and correct pixels in the image. We use Swin model on 200 ADE20K images.

| method | mIoU | big | | all | | %C |
| :--- | :---: | :---: | :---: | :---: | :---: | :---: |
| | | mR | cmR | mR | cmR | |
| Demasked (our) | **19.09** | **66.03** | **52.71** | **23.02** | **12.66** | **47.05** |
| DRS-S | 0.42 | 11.35 | 9.08 | 1.04 | 0.83 | 28.01 |
| DRS-E | 9.12 | 54.67 | 41.78 | 11.04 | 7.86 | 45.03 |

# I  A list of big classes

In Section C we suggest another perspective on the evaluation of our DEMASKED SMOOTHING by specifically considering its performance on "big" semantic classes. The object of these classes occupy on average more than 20% of the images in which they appear. Correctly segmenting these classes is important for understanding the scene. In Tables 9 and 10 we provide the full list of such classes in ADE20K [31] and COCO-Stuff-10K [39] respectively together with the average fraction of pixels that they occupy in the images in which they are present. We point out that for COCO-Stuff-10K some typically smaller classes such as "sandwich" or "fruit" get included in the list of big classes because of the macro-scale images in which they occupy a big part of the scene.

# J  Complexity analysis and parallelization

In DEMASKED SMOOTHING, we propose a set of $K$ masks that are applied to the original image (denote the cost of applying a single masking by $M$). As illustrated in Figure 1c, the masked images are demasked (denote the cost of demasking an image by $D$) and segmented (denote the cost of segmenting an image by $S$); thereupon per-mask segmentations are aggregated into a final segmentation and certification (cost of aggregation proportional to $K$). Asymptotically, compute grows thus with $O(K(M + D + S) + K)$ while the cost of a standard segmentation is $O(S)$. Thus, for large $K$ or $M + D \gg S$, real-time applicability would actually be impractical. However, we note that:

1. $M + D$ is roughly of the same size as $S$ for typical DL-based inpainting and segmentation models.

2. For certified recovery, we operate in a setting where K is small ($K \in 5, 7, 9$) and does not grow with the image resolution. This is unlike Derandomized Smoothing and its derivatives, where the number of masks in the recovery task grows with the image resolution

Table 9: The list of 19 "big" classes for ADE20K [31] (out of 150 classes in total) with their average fraction of occupied pixels in the images where they are present (%) and index in the list of dataset classes. We define a class to be "big" if it occupies on average more than 20% of the pixels in the images in which this class appears.

| # | index | name | fraction | # | index | name | fraction |
|---|-------|------|----------|---|-------|------|----------|
| 1 | 0 | wall | 25.88 | 11 | 79 | hovel | 25.93 |
| 2 | 1 | building | 32.36 | 12 | 88 | booth | 23.91 |
| 3 | 2 | sky | 21.54 | 13 | 96 | escalator | 20.96 |
| 4 | 7 | bed | 21.25 | 14 | 103 | ship | 26.81 |
| 5 | 21 | water | 22.10 | 15 | 104 | fountain | 28.81 |
| 6 | 29 | field | 22.97 | 16 | 107 | washer | 22.07 |
| 7 | 46 | sand | 21.22 | 17 | 109 | swimming pool | 28.87 |
| 8 | 48 | skyscraper | 42.92 | 18 | 114 | tent | 34.57 |
| 9 | 54 | runway | 28.05 | 19 | 128 | lake | 34.57 |
| 10 | 55 | case | 37.57 | | | | |

Table 10: The list of 21 "big" classes for COCO-Stuff-10K [39] (out of 171 classes in total) with their average fraction of occupied pixels in the images where they are present (%) and index in the list of dataset classes. We define a class to be "big" if it occupies on average more than 20% of the pixels in the images in which this class appears.

| # | index | name | fraction | # | index | name | fraction |
|---|-------|------|----------|---|-------|------|----------|
| 1 | 6 | bus | 21.46 | 11 | 105 | floor-stone | 20.10 |
| 2 | 7 | train | 23.11 | 12 | 111 | fruit | 20.48 |
| 3 | 20 | cow | 24.17 | 13 | 113 | grass | 23.25 |
| 4 | 21 | elephant | 28.50 | 14 | 134 | playingfield | 38.64 |
| 5 | 49 | sandwich | 23.99 | 15 | 137 | river | 40.01 |
| 6 | 51 | broccoli | 20.18 | 16 | 143 | sand | 26.37 |
| 7 | 54 | pizza | 25.86 | 17 | 144 | sea | 36.51 |
| 8 | 60 | bed | 36.86 | 18 | 146 | sky-other | 22.94 |
| 9 | 61 | dining table | 21.71 | 19 | 148 | snow | 51.60 |
| 10 | 95 | clouds | 24.11 | 20 | 159 | vegetable | 20.35 |
| | | | | 21 | 167 | water-other | 21.67 |

(or randomized smoothing with thousands of samples per input). This small value of K benefits our the method in time-sensitive applications. For certified detection, we can adjust the number of masks for the computational speed by using strided masking as suggested in Section 4.1.

3. Moreover, masking, demasking, and segmenting for different masks do not use any shared data and can thus be fully parallelized if sufficiently powerful hardware is available. Only the aggregation step requires the results of all the previous stages. However, aggregation time is small compared to the other stages. In terms of latency, a fully parallelized version of our procedure would thus have a latency proportional to $O(M + D + S + K)$. For small $K$ and $M + D \approx S$, application to real-time video can be facilitated by means of parallelization.

# K   Used data

In this work, we only use the datasets published under formal licenses: ADE20K [31] and COCO-Stuff-10K [39]. To the best of our knowledge, data used in this project do not contain any personally identifiable information or offensive content. The models ZITS [29] and Swin [26] are published under Apache-2.0 license. The text of the license for PSPNet [24] can be found here: `https://github.com/hszhao/PSPNet/blob/master/LICENSE`

Table 11: Execution time comparison for the BEiT-B model on 2000 ADE20K images. Vanilla segmentation average per-image run time is 387 miliseconds. Computations were done on a single Nvidia Tesla V100 GPU. We provide total Demasked Smoothing time as well as the execution time of different stages of the method. We use $K = 20$ masks for detection and $K = 5, 7, 9$ masks for recovery with $T = 2, 3, 4$ respectively.

| dataset | segm | mode | mask type | average per-image run time (ms) | | | | | overhead |
| | | | | mask | demask | segment | aggregate | total | |
|---|---|---|---|---|---|---|---|---|---|
| ADE20K | BEiT-B | detection 1% patch | column | 428 | 9125 | 8429 | 352 | 18334 | ×47 |
| | | | row | 251 | 9485 | 8503 | 348 | 18587 | ×48 |
| | | recovery 0.5% patch | column | 376 | 3439 | 2446 | 563 | 6824 | ×18 |
| | | | row | 194 | 3329 | 2460 | 645 | 6628 | ×17 |
| | | | 3-mask | 194 | 4370 | 3319 | 750 | 9025 | ×22 |
| | | | 4-mask | 243 | 5020 | 4063 | 825 | 10151 | ×26 |

## L   Demasked Smoothing Visualization

In this section, we provide additional illustrations of our method (Figures 14, 15, 16, 17). Similarly to the Table 1 we certify against a 1% patch for the detection task and against 0.5% patch for the recovery task. For each mask type we illustrate all the stages summarized in the Figure 1c. We also provide examples of certification maps for certified recovery and certified detection with different images (Figure 18, 19).

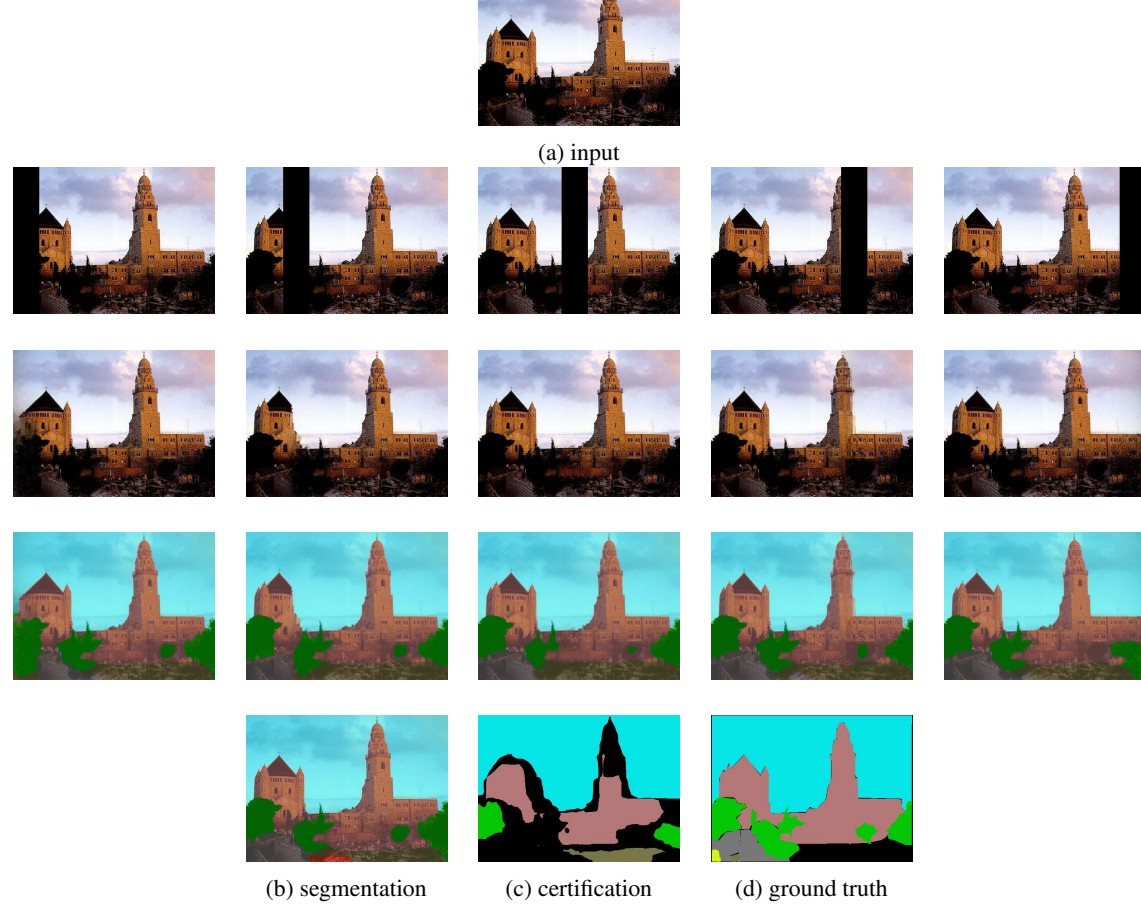

(a) input

(b) segmentation  (c) certification  (d) ground truth

Figure 14: DEMASKED SMOOTHING detection column masking illustration for an image from ADE20K [31]. We illustrate five masks out of twenty.

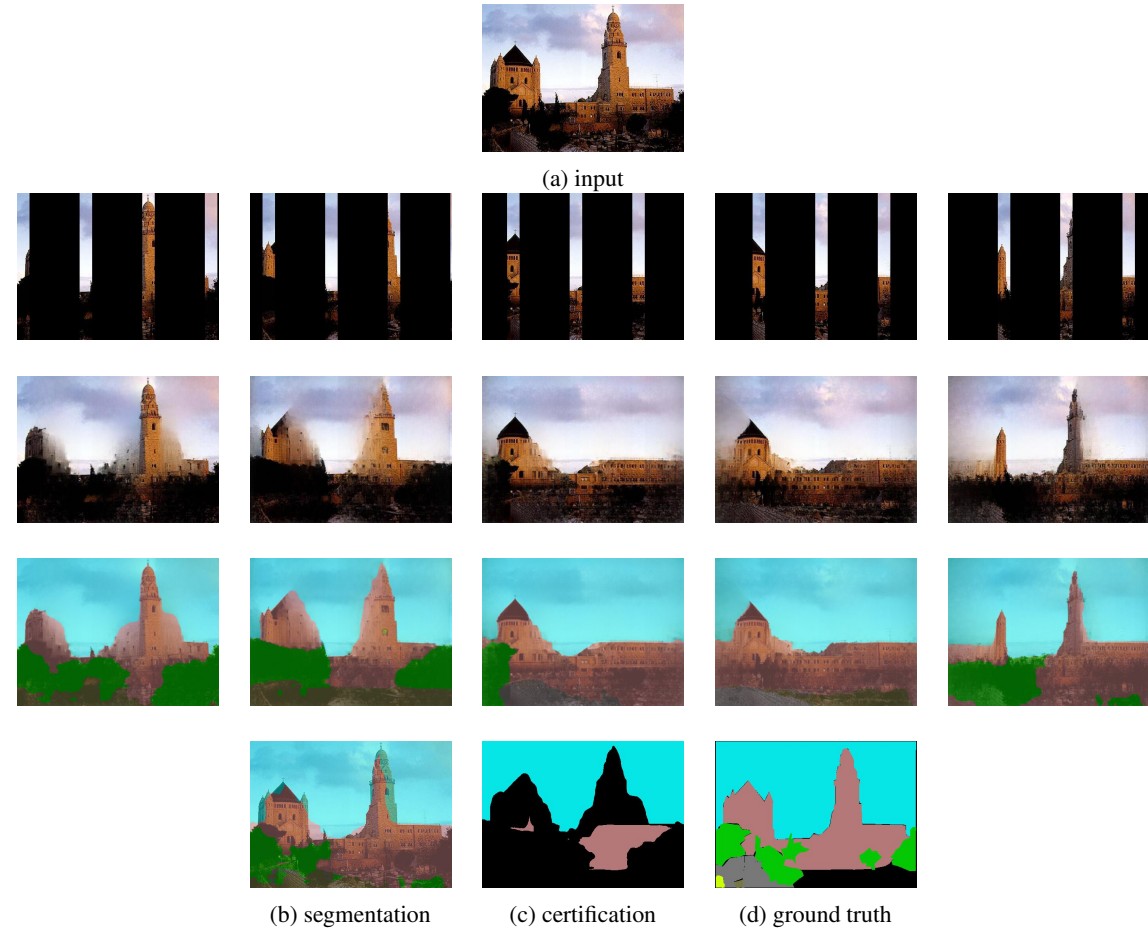

(a) input

(b) segmentation     (c) certification     (d) ground truth

Figure 15: DEMASKED SMOOTHING recovery column masking illustration for an image from ADE20K [31].

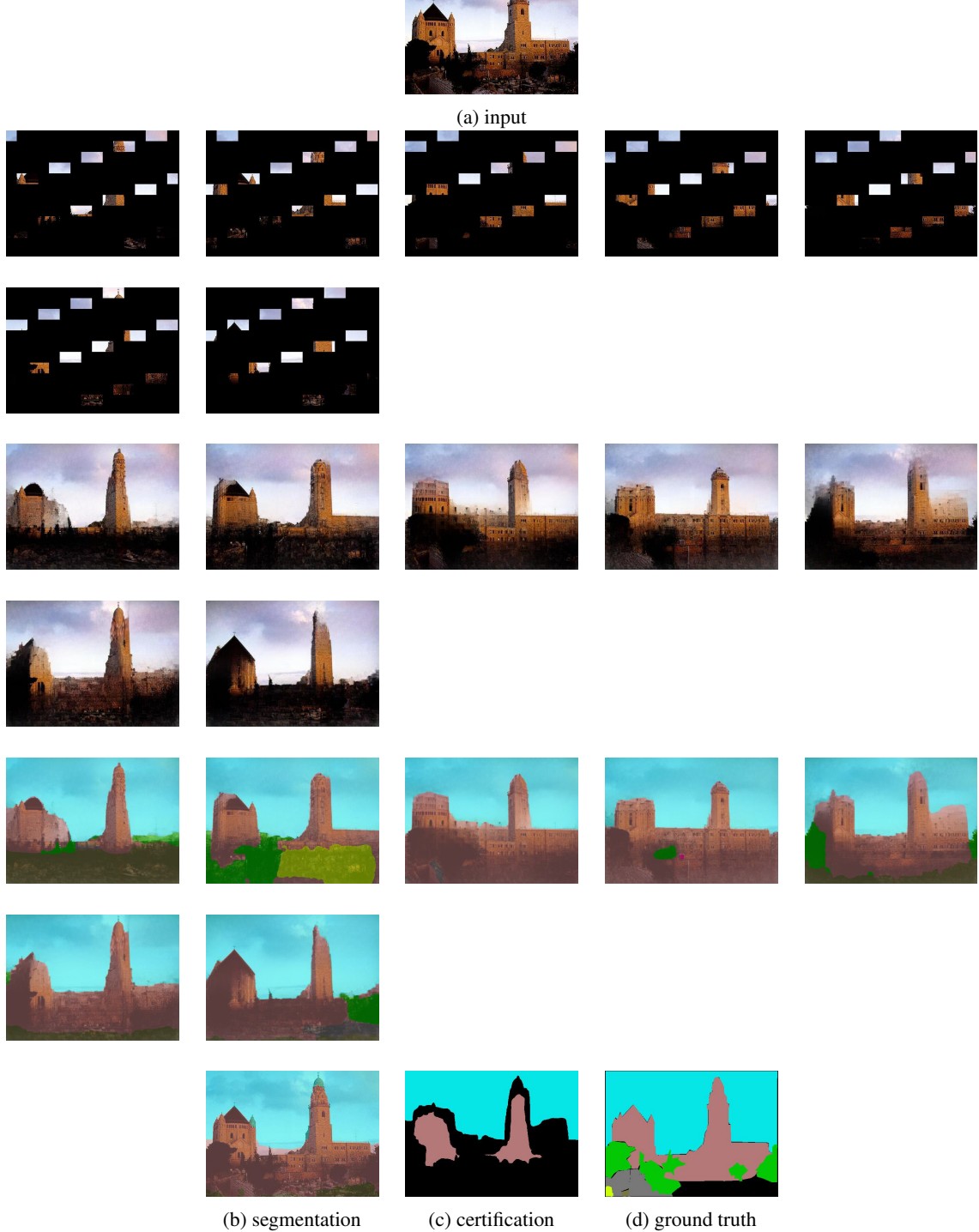

(a) input

(b) segmentation  (c) certification  (d) ground truth

Figure 16: DEMASKED SMOOTHING recovery masking for $T = 3$, $K = 7$ masks (Section 4.1) illustration for an image from ADE20K.

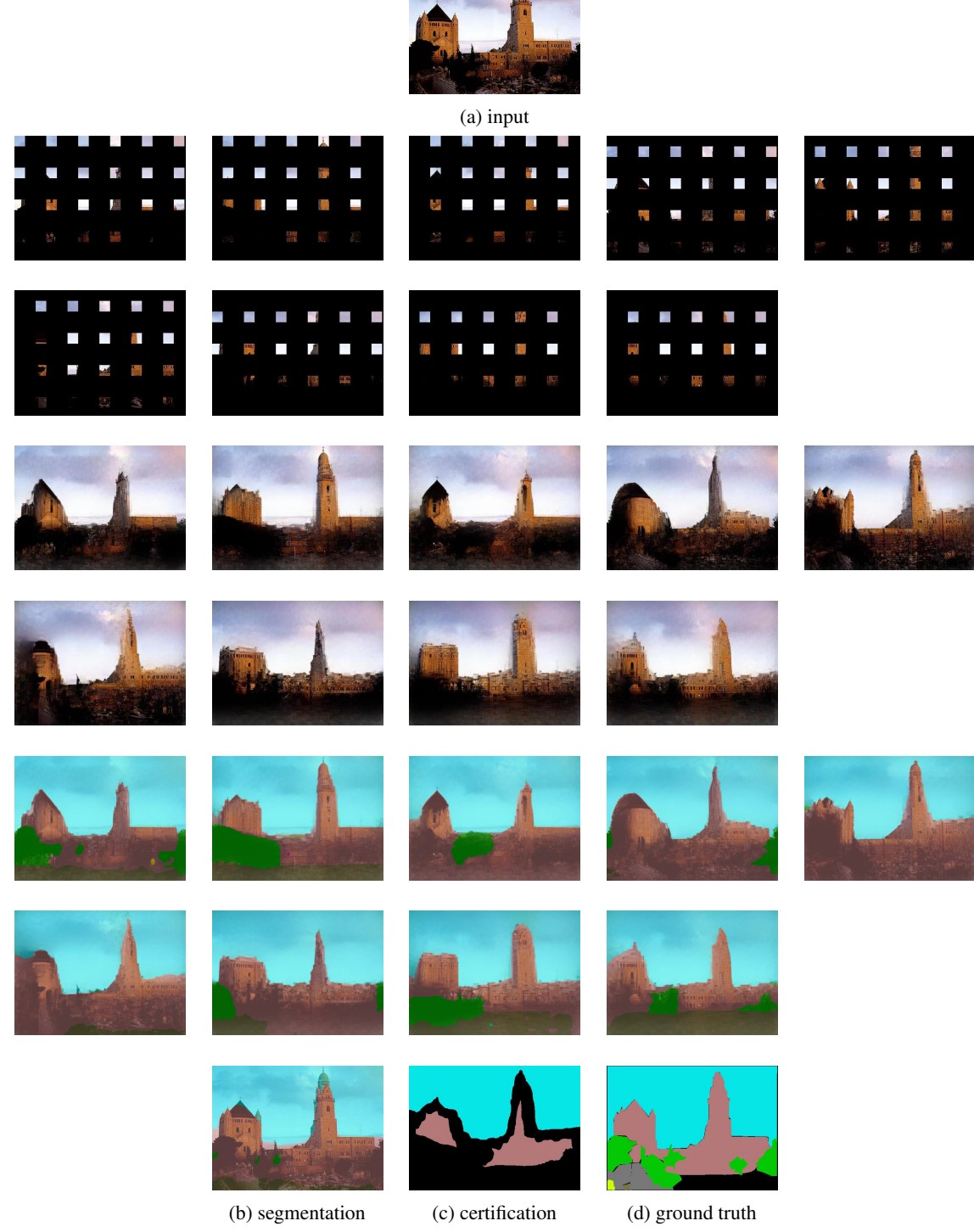

(a) input

(b) segmentation  (c) certification  (d) ground truth

Figure 17: DEMASKED SMOOTHING recovery masking for $T = 4$, $K = 9$ masks (Section 4.1) illustration for an image from ADE20K [31].

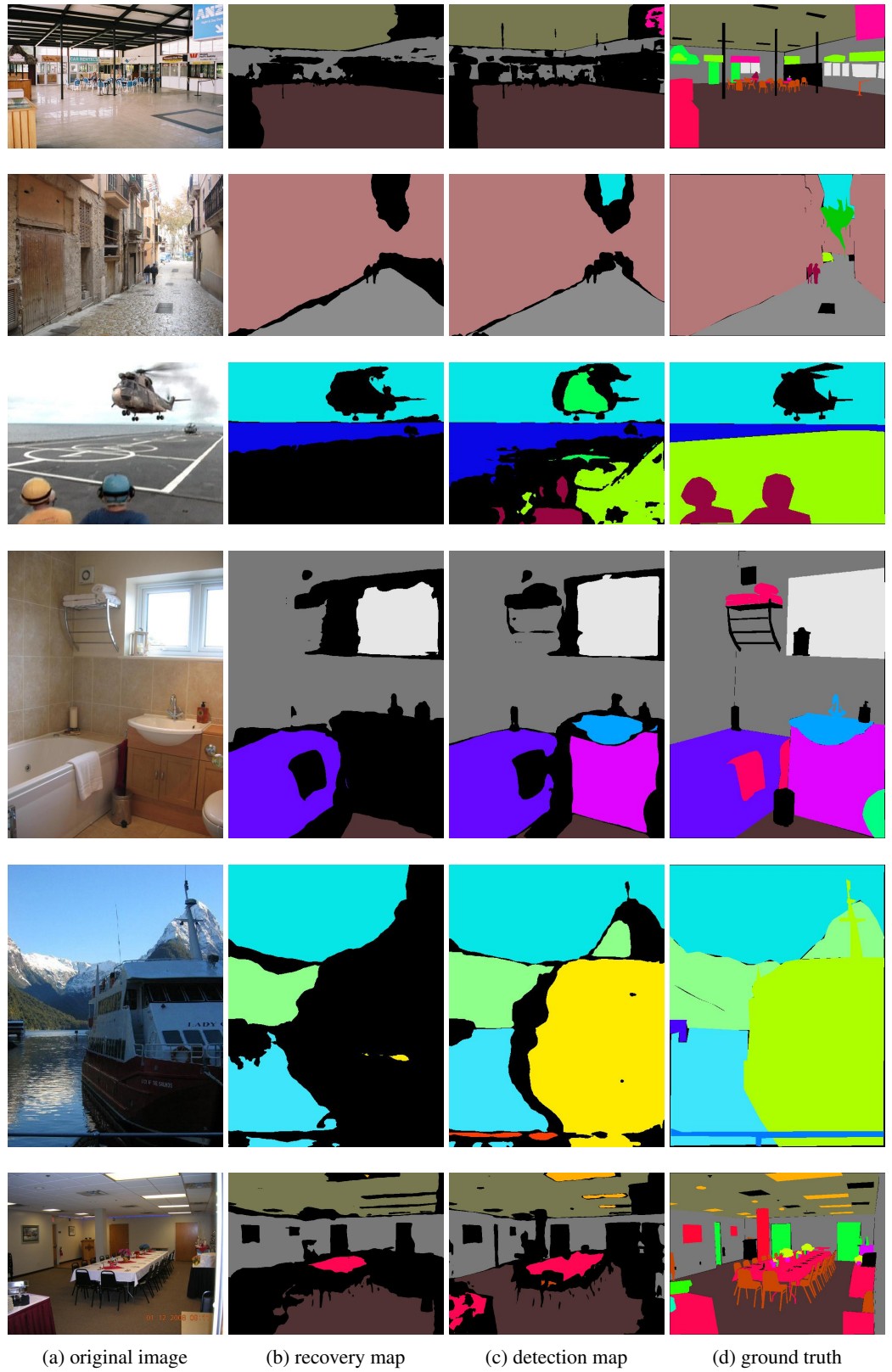

(a) original image      (b) recovery map      (c) detection map      (d) ground truth

Figure 18: Certification map examples on ADE20K [31] with ZITS [29] and Swin [26].

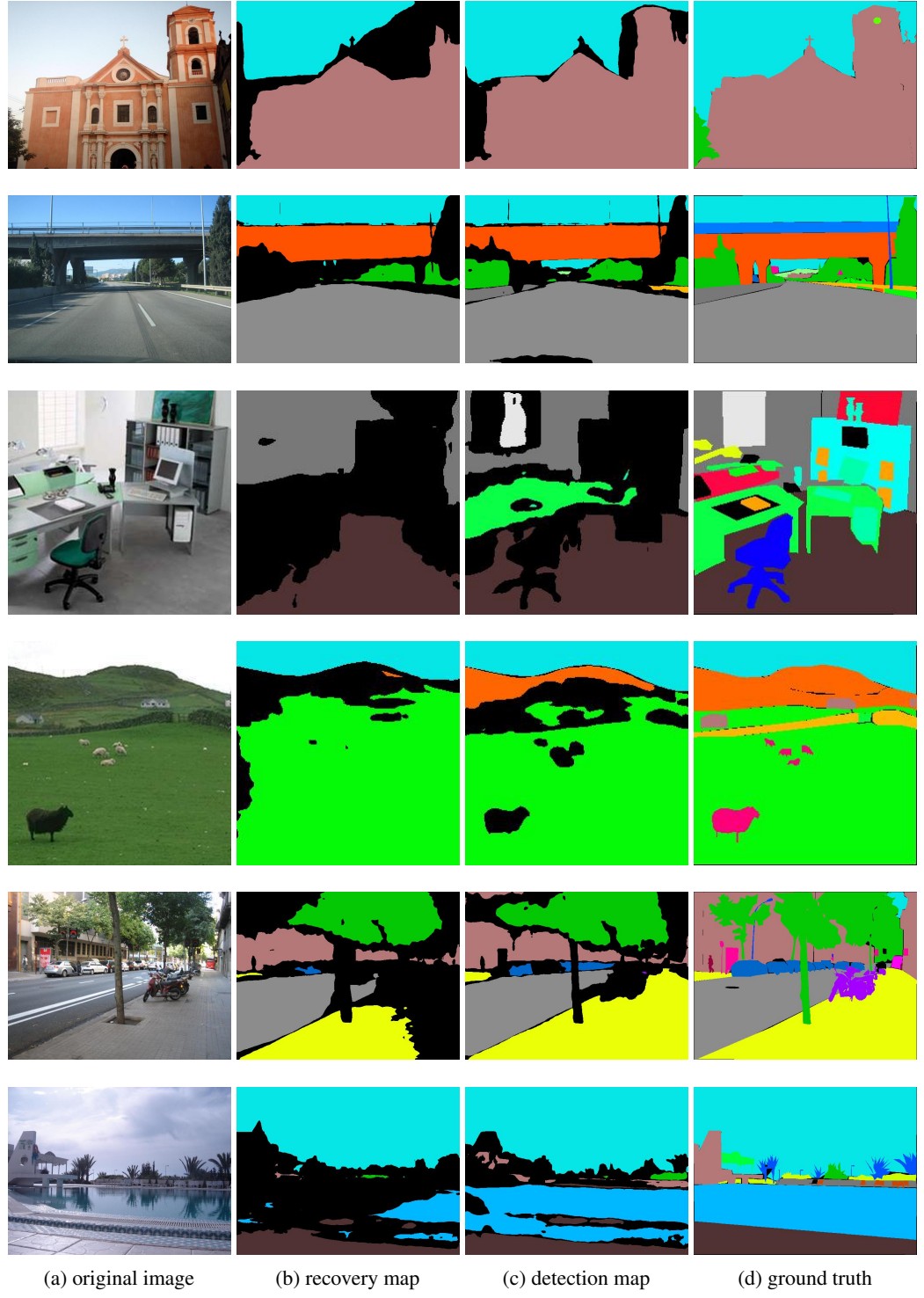

(a) original image      (b) recovery map      (c) detection map      (d) ground truth

Figure 19: Certification map examples on ADE20K [31] with ZITS [29] and Swin [26].

