# OpenReview forum: "Certified Defences Against Adversarial Patch Attacks on Semantic Segmentation"
_NeurIPS.cc/2022/Workshop/TSRML — TSRML2022_

### Official Review · Reviewer_1YEm · 2022-10-10
**Extension of existing ideas on certified robustness to adversarial patches to Semantic Segmentation**

**Overall Rating:** 6

**Summary:**

This paper takes an existing idea to create certified robustness to adversarial patch attacks: generate several version of the image (here done by masking, but can be also found as cropping or dropping patches in other works), evaluate the model on those multiple version, and take a majority vote over the final prediction. If the vote are unanimous, then because the patch is dropped in most of the versions of the image, it can only impact a few of them, so it has no possibility to reverse the voting decision.

The novelty of this paper lies in the application domain, semantic segmentation. This is problematic because a prediction needs to be made for every pixel, which makes things complicated when the corresponding pixels are dropped. The proposed solution relies on in-painting to reconstruct a full image, and applying the segmentation model to these reconstructed images. This allows the proposed method to not require re-training.

**Strengths:**

The authors do a good job at explaining their method and the figures are helpful to grasp an intuitive understanding of the procedure. Parsing the theorems and all the notation requires a bit of work but is in the end relatively ok.

To the best of my knowledge, the application of these masking-and-voting method to semantic segmentation is novel, and the solution proposed by the author to make it work is quite simple and easy to apply, although I have my reservations about the meaningfulness of the task and the evaluation, discussed below.

The empirical evaluation is thorough, testing over several datasets, applying their technique to several segmentation method, and performing  several ablation of the components of their system.

**Weaknesses:**

# Questions about experimental results.
- Why is the data omitted when row masks are used for detection in Table 1?
- Lacking from the data reported is a discussion of the impact on evaluation time of the method. If a single pass through a segmentation method is replaced by K rounds of inpainting + segmentation, this should be relatively significant and therefore deserve to be noted.

# Applicability of the evaluation
I'm a little bit unsure on how to appreciate the results that are given. Giving data like %C, that corresponds to proportion of correct pixels in the image,  is going to be quite biased towards a method that is really good at certifying large classes (like the sky or the road in a driving scenario) while ignoring all details (car in the distance, pedestrians).  This is essentially what I'm observing in Figure 9 and 10: Sky is fine, giant building is fine, road is fine, but the model loses most of its ability to detect stuff like pedestrians, which are really important, but also, if they are masked, are going to be really prone to being removed when inpainting happens. I understand that this is a limitation of the task as well (if there is a patch masking a pedestrian, you can't detect it) but in that case, I wonder if the setup of the problem is a bit  too artificial. Maybe what should be considered is the robustness of the segmentation model _outside of the patch_ ?

# Comparison to related work:
I think it would be beneficial to include some additional discussion of related work with similar mechanism. One of the main idea in this paper is the use of masking + majority voting to be able to provide some guarantees, which can be found in other works (such as [13] which I could not find a discussion of in the body of the paper, [9] or [15]). There is still valid contribution in this paper (the addition of in-painting in order to make the base idea work with semantic segmentation) but I feel like being more clear in idea attribution would be great. The Related Work section on Certified Recovery should be used to highlight what is similar and what is different, rather than listing that other paper just _exists_. Highlight the common parts and the difference.

# Minor comments:
- Typo "tranfromer" l.41
- L.77-78 -> In the definition of Certified detection, do you mean to say that the detection method is considered successful as long as it detects that there was an adversarial patch present, even if the prediction of the model is affected? As it is, the sentence is a bit ambiguous and sounds like it's acceptable to have the model prediction accepted and that if the model prediction is affected, it means that the attack is detected.
- In theorem 1, what does "for all S[k] \in S" means? Is it "for all k in [1...K]", to mean that you are enumerating over the results of the segmentation for each mask?
- It might be useful to disambiguate a bit between the definition of Certified Detection (l. 109 to 111) and Theorem 2. My initial reaction was that this was a direct application of the definition but maybe it would be possible to rewrite it as defining the function over the masked/demasked segmentation as v_hat, and then make it clear that the point of Theorem 2 is to say "v_hat is a valid certified detection function as it satisfies the definition we introduced above."
- Typo "patchn" l.184
- Typo "horizont" l.195
- Typo "certificaton" l.421




**Overall Recommendation:**

The paper is interesting to read but would benefit from better discussion of related work (for contextualizing the novelty) and of the weaknesses of the solution (higher runtime, lack of sensitivity to small objects). It is still however worth beneficial to have a reference point given for what is achievable for segmentation model under patch attacks.

**Review Confidence:**

4: The reviewer is confident but not absolutely certain that the evaluation is correct

---

### Official Review · Reviewer_BtXS · 2022-10-11
**Reivew**

**Overall Rating:** 8

**Summary:**

The paper proposed a Demasked Smoothing defense for certifiable robustness of image segmentation against adversarial patches. The defense applies masks to the input, uses image inpainting to recover the masked pixels, and checks prediction consistency or majority predictions for the final predictions.

**Strengths:**

1. First to certifiably robust defense for semantic segmentation against adversarial patches
2. the algorithm is compatible with different segmentation models


**Weaknesses:**

1. The defense performs poorly when there is no attack, which would greatly discourage its use in practice
    1. the detection-based defense has a very high false alert rate in the clean setting (e.g., 20%).
    2. the recovery-based defense has a large drop in clean mIoU (e.g., 50%->24%)
2. the defense overhead is very large. The defense needs to do inpainting and inference on many masked images.
3. only have guarantees for recall, but not precision


**Overall Recommendation:**

The paper proposed an effective certifiably robust defense against adversarial patches. It is the first defense for the segmentation tasks, and also achieves non-trivial certifiable robustness. I believe its contributions and strengths outweigh its weaknesses of large overhead and poor clean performance.

**Review Confidence:**

5: The reviewer is absolutely certain that the evaluation is correct and very familiar with the relevant literature

---

### Official Review · Reviewer_VkaP · 2022-10-20
**Review: Certified Defences Against Adversarial Patch Attacks on Semantic Segmentation**

**Overall Rating:** 8

**Summary:**

This paper proposes the first certified defense against adversarial patches for the task of image segmentation. Their proposed method can work with any off-the-shelf segmentaion model, and therefore is a general-purpose solution.

The propose method involves: (1) Generating several masked copies of a given image; (2) Unmasking the masked copies using an off-the-shelf inpainting mode; (3) passing the inpainted images through the segmentation model and aggregating outputs.

This process of aggregating results over masked copies yields certified robustness against patch-based adversarial attacks.

**Strengths:**

- The proposed solution is general-purpose, i.e., it can work with any off-the-shelf segmentaiton model. The proposed method is conceptually and technically sound.
- The proposed solution can be used for certified recovery as well as detection.
- The paper is well written and the mathematical formulations have been gracefully presented, i.e., they are easy to follow. The theorectical components have been adequately discussed.
- The empirical section is thorough and appropriately highlights the contributions of the paper.

**Weaknesses:**

The novelty of the proposed approach is somewhat incremental. The overall framework of using a masking mechanism to obtain certified robustness against adversarial patches has been previously explored in context of image classification and object detection (as acknowledged by the author). The authors use the same framework, but adopt it to the image segmentation domain by adding an inpainting step. Therefore, the only novel component they introduce is the addition of the inpainting step.

**Overall Recommendation:**

This paper proposes a novel application of an existing strategy (with some modifications). I believe the contributions made by the paper are meaningful enough for acceptance in this workshop.

**Review Confidence:**

3: The reviewer is fairly confident that the evaluation is correct

---

### Decision · Program_Chairs · 2022-10-23

Accept